# Do Individualized Patient-Specific Situations Predict the Progression Rate and Fate of Knee Osteoarthritis? Prediction of Knee Osteoarthritis

**DOI:** 10.3390/jcm12031204

**Published:** 2023-02-02

**Authors:** Hyun Jin Yoo, Ho Won Jeong, Sung Bae Park, Seung Jae Shim, Hee Seung Nam, Yong Seuk Lee

**Affiliations:** 1Department of Orthopaedic Surgery, Seoul National University College of Medicine, Seoul National University Bundang Hospital, Seoul 13620, Republic of Korea; 2Department of Orthopedic Surgery, Konyang University College of Medicine, Daejeon 35365, Republic of Korea

**Keywords:** knee, patient-specific data, osteoarthritis, healthcare, progression rate of osteoarthritis, fate of osteoarthritis

## Abstract

Factors affecting the progression rate and fate of osteoarthritis need to be analyzed when considering patient-specific situation. This study aimed to identify the rate of remarkable progression and fate of primary knee osteoarthritis based on patient-specific situations. Between May 2003 and May 2019, 83,280 patients with knee pain were recruited for this study from the clinical data warehouse. Finally, 2492 knees with pain that were followed up for more than one year were analyzed. For analyzing affecting factors, patient-specific information was categorized and classified as demographic, radiologic, social, comorbidity disorders, and surgical intervention data. The degree of contribution of factors to the progression rate and the fate of osteoarthritis was analyzed. Bone mineral density (BMD), Kellgren–Lawrence (K–L) grade, and physical occupational demands were major contributors to the progression rate of osteoarthritis. Hypertension, initial K–L grade, and physical occupational demands were major contributors to the outcome of osteoarthritis. The progression rate and fate of osteoarthritis were mostly affected by the initial K–L grade and physical occupational demands. Patients who underwent surgical intervention for less than five years had the highest proportion of initial K–L grade 2 (49.0%) and occupations with high physical demand (41.3%). In identifying several contributing factors, the initial K–L grade and physical occupational demands were the most important factors. BMD and hypertension were also major contributors to the progression and fate of osteoarthritis, and the degree of contribution was lower compared to the two major factors.

## 1. Introduction

Osteoarthritis (OA) represents an important healthcare problem in an aging society because its incidence increases with aging. OA involves inflammation and major structural changes in the joints [1]. This results in irreversible damage to the joint cartilage and bony structures [2]. The knee joint is the most common site of OA, and it causes physical disability and chronic pain [3,4]. Globally, over 250 million people have symptomatic knee OA. Furthermore, the prevalence of symptomatic knee OA is particularly higher among Asians [5]. Despite the high incidence of knee OA, few drugs have been proven to effectively modify disease progression, and only a few treatment options have been proven to relieve symptoms. Considering the failure to find an effective disease-modifying OA drug (DMOAD) or nonsurgical treatment, it is necessary to prevent or delay the degenerative process in the absence of radiologic or mild OA with knee pain [6,7].

OA has a multifactorial etiology, which occurs because of the interaction between systemic and local factors. Some meta-analyses have shown that hypertension and diabetes mellitus are highly related with OA [8,9]. Moreover, there was strong evidence about greater risk in females for all stages of knee OA. Females also tended to have more severe knee OA than males, and this tendency was more prominent among people aged over 55 years [10,11]. Furthermore, bilateral involvement of knee OA was more prevalent in women [11]. Therefore, factors affecting the progression rate and fate of primary OA need to be analyzed while considering patient-specific situations, such as demographic, radiologic, social, and comorbidity disorders data.

OA is usually diagnosed and studied based on a combination of demographics, clinical history, physical examination, and radiologic imaging [12,13,14]. Despite the current effective diagnostic modalities and availability of studies on factors for OA, the characteristics of patients who progress more rapidly and eventually require surgical intervention are not known. It is important to predict the progression rate because patients who are at risk of rapid progression require active management [15]. It is also important to predict the fate of OA, because long-term planning is then possible in the primary healthcare system. However, most studies on OA address heterogeneous OA grades and are not specific to individual patients.

This study aimed to identify and provide information about the predicted progression rate and the fate of OA based on patient-specific situations such as demographic, radiologic, social, and comorbidity disorders data. We also focused on early OA (Kellgren–Lawrence [K–L] grade 0–2) because long-term management plans were more important at this stage. Therefore, the purpose of this study was to identify the rate of remarkable progression and fate of primary OA in patient-specific situations. This study hypothesized that the progression rate and fate of OA would differ according to patient-specific situations, even for patients with a similar grade of OA at the initial screening.

## 2. Materials and Methods

Institutional Review Board approval was obtained before performing the study. Between May 2003 and May 2019, 83,280 patients with knee pain were recruited for this study. Consistent data obtained from the clinical data warehouse (CDW) of a single institution were utilized accordingly. For patient-specific situations, data were categorized as follows: (1) demographic data, including age, sex, body mass index (BMI), and BMD; (2) radiological data including official readings of initial and final radiographs of the knee joint (the final radiograph was defined as the time of the last K–L grade or just before surgery); (3) social data including occupation; (4) comorbidity data including hypertension, diabetes mellitus, and others (tuberculosis, liver disease, cardiovascular disease, cancer, epilepsy, and kidney disease); and (5) surgical data including information about surgical interventions. The exclusion criteria were as follows: (1) patients who started follow-up for less than 10 years from the time of data acquisition (accordingly, patients who had visited before at least May 2010 were enrolled); (2) a conservative treatment period of less than one year; (3) initial K–L grade 3 or 4; and (4) history of lower extremity diagnosed with major trauma, including fracture, ligament, cartilage, and meniscal injury by the event. We excluded secondary OA for the prediction of primary OA progression. Finally, a total of 2492 knee data points (1678 patient data) were enrolled in this study.

### 2.1. Classification

Age was divided into three categories for even distribution (age < 55 years, 55 ≤ age < 65 years, and age ≥ 65 years). These divided categories of age were used for comparing according to K–L grade, the progression rate, and the fate of OA. The earliest BMD value was used for analysis, and when not performed, patients were considered to have missing values. BMD was divided into two categories: normal (T-score ≥ −1.0) and osteopenia or osteoporosis (T-score < −1.0). To check the remarkable change, the progression rate of OA was defined as the time elapsed for two or more K–L grade changes from no radiologic or mild OA.

The progression rate of OA was divided into three groups for even distribution (fast: time < 5 years; usual: 5 years ≤ time < 10 years; and slow: time ≥ 10 years). In terms of physical demand, occupation was classified based on the degree of kneeling, squatting, lifting, walking, climbing, and standing while referencing the relative risk of occupation to OA, as described by Palmer et al. [14,16]. We divided the physical demands of occupations into three categories: (1) low demand: white-collar workers (office workers), students, and jobless individuals; (2) mild demand: workers in the manufacturing industry, kitchen and serving staff, soldiers, and light manual workers; and (3) high demand: construction workers, factory workers, farmers, professional sports players, and hard laborers. Three categories of comorbidity factors were selected for analysis considering incidence: (1) hypertension, (2) diabetes mellitus, and (3) other disorders (one or more other comorbidities). The outcome of OA was defined as surgical or nonsurgical treatment. Osteotomy, unicompartmental knee arthroplasty (UKA), and total knee arthroplasty (TKA) were included as surgical interventions for knee OA. The fate of OA was classified as an elapsed time to surgical intervention. The elapsed time to surgical intervention was divided into two groups based on the balance in the data (five-year basis). If there was no surgical intervention, it was included in the conservative treatment group. Descriptions were summarized in Table 1.

### 2.2. Affecting Factor Analysis

The analysis was built by a model for predicting the progression rate and the fate of OA. The degree of contribution of factors to the progression rate and the fate of OA was analyzed. The Scikit-learn library 0.23.2 in Python (version 3.6; Python Software Foundation, Wilmington, DE, USA) was used for our prediction model (Figure 1). We used multinomial logistic regression for predicting the rate of progression and fate of OA. SoftMax was used as a classifier of multinomial logistic regression and cross-entropy loss was used for calculating error at training (difference of SoftMax with the true label, Appendix A).

For validation of the model, data were randomly partitioned into 70%:30%-sized groups during each iteration to ensure independence between the training data and test data. Furthermore, we iterated 100 times to analyze the validity of the models by calculating the 95% confidence intervals. Age and BMI were used as continuous variables of the prediction model. The categorical variables of the prediction model were classified as follows: (1) sex: male = 0, female = 1; (2) BMD: normal = 0, osteopenia or osteoporosis = 1; (3) physical occupational demands: low = 0, mild = 1, high = 2; and (4) comorbidities: absent = 0, present = 1. Confusion matrices were used to calculate the performance of the model (Figure 2). Indicating numbers of Figure 2 were as follow: (a) time of K–L grade change more than 2, 0: time ≥10 years, 1: 5 years ≤ Time <10 years, 2: time <5 years, (b) time of surgical intervention, 0: conservative, 1: surgical intervention ≤5 years, 2: surgical intervention >5 years (maximum: 15 years).

The performance of the model was assessed by accuracy, precision, recall, F1-score, specificity, and error rate. The prediction model was built by using factors in Table 2 and Table 3 and used to obtain the coefficient for estimating the effects of each factor. Coefficients were classified into two ranks by calculating the average of all coefficients of each category to determine the relative effect (average coefficient as a major contributor ≥ |0.211| and average coefficient as a minor contributor < |0.211|).

### 2.3. Statistical Analysis

Conventional statistical analyses were performed by using SPSS version 18.0. (IBM Corp., Armonk, NY, USA). Data descriptions were based on the means and standard deviations for continuous variables. The chi-square test was used to compare categorical variables (sex, BMD, occupation, and comorbidities). Analysis of variance (ANOVA) was performed to compare age, initial and final K–L grade, and BMI between the study groups. Tukey’s honestly significant difference (Tukey HSD) was used for post-hoc analysis. The results were considered statistically significant when the *p* value was <0.05. The conventional statistical analysis was used to compare each factor according to K–L grade (Table 1), the progression rate (Table 2), and the fate of OA (Table 3).

**Table 2 jcm-12-01204-t002:** Comparison of affecting factors according to the progression rate of OA.

Total (Knees)	Time < 5 Years (Fast)	5 years ≤ Time < 10 Years (Usual)	Time ≥ 10 Years (Slow)	*p*-Value
**100%** (2152)	36.0% (774)	33.5% (721)	30.5% (657)	
**Age** (Mean)	62.2 ± 8.8	58.6 ± 9.2	58.9 ± 11.8	<0.001 *
**Age < 55** (553)	24.2% (134)	38.5% (213)	37.3% (206)	<0.001 *
**55 ≤ Age < 65** (894)	36.4% (325)	36.7% (328)	27.0% (241)	
**Age ≥ 65** (705)	44.7% (315)	25.5% (180)	29.8% (210)	
**SEX** (Male/Female)	12.7%/87.3% (98/676)	18.6%/81.4% (134/587)	19.6%/80.4% (129/528)	
**BMI**	23.8 ± 3.4	25.0 ± 3.2	25.4 ± 3.3	<0.001 *
**BMD**				<0.001 *
**Normal** (134)	39.6% (53)	32.1% (43)	28.4% (38)	
**Osteopenia or Osteoporosis** (553)	43.6% (241)	32.2% (178)	24.2% (134)	
**Total performed** (687)	42.8% (294)	32.2% (221)	25.0% (172)	
**Initial K–L grade**				<0.001 *
**K–L grade 0** (1542)	29.4% (453)	37.7% (581)	32.9% (508)	
**K–L grade 1** (296)	38.5% (114)	27.4% (81)	34.1% (101)	
**K–L grade 2** (314)	65.9% (207)	18.8% (59)	15.3% (48)	
**Physical demand for occupation**				<0.001 *
**Low demand** (719)	32.0% (230)	26.7% (192)	41.3% (297)	
**Mild demand** (914)	32.1% (293)	38.7% (354)	29.2% (267)	
**High demand** (519)	48.4% (251)	33.7% (175)	17.9% (93)	
**Metabolic disorders**				
**HTN** (1091)	40.8% (445)	31.7% (346)	27.5% (300)	<0.001 *
**DM** (405)	39.8% (161)	31.9% (129)	28.4% (115)	0.208
**Other disorders** **(one or more)** (517)	35.4% (183)	35.0% (181)	29.6% (153)	0.699

K–L, Kellgren–Lawrence; time, elapsed K–L grade change more than 2; OA, osteoarthritis; BMI, body mass index; BMD, bone mineral density; HTN, hypertension; DM, diabetes mellitus. * Statistical significance was *p* < 0.05.

**Table 3 jcm-12-01204-t003:** Comparison of affecting factors according to the fate of OA.

(Knees)	Time < 5 Years	Time ≥ 5 Years (Max: 15 Years)	Conservative Treatment	*p*-Value
**Age** (Mean)	63.5 ± 7.0	59.6 ± 9.3	56.9 ± 11.9	<0.001 *
**SEX** (Male/Female)	13.5%/86.5% (42/270)	12.8%/87.2% (56/383)	21.0%/79.0% (366/1375)	<0.001 *
**BMI**	23.9 ± 3.4	24.0 ± 3.2	25.2 ± 3.4	<0.001 *
**BMD**				<0.001 *
**Normal** (159)	9.4% (14)	14.0% (25)	27.8% (120)	
**Osteopenia or osteoporosis** (600)	90.6% (135)	86.0% (154)	72.2% (311)	
**Total performed** (759)	100% (149)	100% (179)	100% (431)	
**Initial K–L grade**				<0.001 *
**K–L grade 0** (1850)	27.5% (86)	43.3% (190)	86.6% (1507)	
**K–L grade 1** (303)	23.5% (73)	18.9% (83)	10.2% (178)	
**K–L grade 2** (339)	49.0% (153)	37.8% (166)	3.2% (56)	
**Total** (2492)	100% (312)	100% (439)	100% (1741)	
**Physical demand for occupation**				<0.001 *
**Low demand** (813)	32.1% (100)	23.2% (102)	35.1% (611)	
**Mild demand** (1109)	26.6% (83)	41.7% (183)	48.4% (843)	
**High demand** (570)	41.3% (129)	35.1% (154)	16.5% (287)	
**Total** (2492)	100% (312)	100% (439)	100% (1741)	
**Rate of metabolic disorders**				
**HTN** (1204)	61.5% (192)	57.2% (251)	43.7% (761)	<0.001 *
**DM** (443)	25.3% (79)	18.7% (82)	16.2% (282)	<0.001 *
**Other disorders** **(one or more)** (575)	16.5% (95)	19.7% (113)	63.8% (367)	0.001 *

OA, osteoarthritis; K–L, Kellgren–Lawrence; BMI, body mass index; BMD, bone mineral density; HTN, hypertension; DM, diabetes mellitus. * Statistical significance was *p* < 0.05.

## 3. Results

Patient demographics were arranged according to the initial K–L grade of no radiologic or mild OA (Table 1). The mean age was the lowest in the initial K–L grade 0 group (57.6 ± 11.9, post hoc *p* < 0.001), and BMI was the lowest in the initial K–L grade 2 group (23.0 ± 2.8, post hoc *p* < 0.001). The initial K–L grade 2 group had the highest prevalence of osteopenia or osteoporosis (*p* < 0.001). The proportion of individuals with high physical demand occupations was the highest in the initial K–L grade 2 group (*p* < 0.001). The rate of comorbidity was the highest in the initial K–L grade 2 group. The highest K–L grade (3.8 ± 0.4) observed at the final follow-up visit was in the initial K–L grade 2 group (post hoc *p* < 0.001).

Comparison of affecting factors according to the progression rate of OA is summarized in Table 2. The fast-progression OA group tended to have a higher proportion of elderly individuals and a higher initial K–L grade (*p* < 0.001). The time required to proceed to the next K–L grade decreased as KL grade increased (Figure 3). Patient-specific data of Figure 3 were as follow: sex, male; age, 51; BMI, 28; occupation, manufacturer (mild physical demand); and comorbidity, not applicable. The average rates of progression from one K–L grade to another were: K–L grade 0 to 1, 10.05 years; K–L grade 1 to 2, 7.29 years; K–L grade 2 to 3, 5.26 years; and K–L grade 3 to 4, 4.15 years. BMI was the lowest in the fast-progression OA group (23.0 ± 2.8, post hoc *p* < 0.001), and osteopenia or osteoporosis was the most prevalent in the fast-progression OA group (*p* < 0.001). Patients in the fast-progression OA group tended to have a higher physical demand for occupation (*p* < 0.001) and a higher incidence of comorbidities, especially hypertension (*p* < 0.001).

A detailed comparison of the outcomes of OA is presented in Table 3. Surgical intervention was performed on the knees within five years in 12.5% (312) of cases, more than five years in 17.6% (439) of cases, and conservative treatment was performed (not undergoing surgical intervention) in 69.9% (1741) of the cases. The mean age of patients who underwent surgical intervention was the highest in the within-five-years group (63.5 ± 7.0, post-hoc *p* < 0.001). Furthermore, participants who underwent surgical intervention within five years had the highest proportion of initial K–L grade 2 (49.0%) (153). Surgical intervention within five years was more frequent in the high physical demand occupation category (*p* < 0.001) and in patients with hypertension or diabetes mellitus (*p* < 0.001).

The detailed performance of the model is summarized in Figure 2 and Table 4. The accuracy of the model for the progression rate of OA was 0.632, 0.616, and 0.644 for fast, usual, and slow progression, respectively, and the accuracies for the fate of OA were 0.874, 0.803, and 0.876 for the within-five-years, more-than-five-years, and conservative groups, respectively. A detailed comparison of the coefficients used to estimate the effect of each factor is presented in Table 5. Initial K–L grade and physical occupational demands were major contributors to both the progression rate and the fate of OA. BMD was the major contributor to the progression of OA. Hypertension was a major contributor to the outcome, but not the progression of OA. The progression rate and fate of OA differed according to the initial K–L grades, even in patients with similar conditions (Figure 4A,B). Patient-specific data of Figure 4A were as follows: (a) sex, female; age, 56; BMI, 25; occupation, no work (low physical demand); and comorbidity, not applicable; (b) sex, female; age, 59; BMI, 27; occupation, no work (low physical demand); and comorbidity, not applicable. Patient-specific data of Figure 4B were as follows: sex, female; age, 64; BMI, 29; occupation, serving worker (mild physical demand); and comorbidity, hypertension.

Among patients with the same initial K–L grades, the progression rate and fate of OA differed according to patient-specific situations, especially physical occupational demands (Figure 5A,B). Patient-specific data of Figure 5A were as follows: (a) sex, male; age, 48; BMI, 26; occupation, office worker (low physical demand); and comorbidity, not applicable; (b) sex, male; age, 46; BMI, 28; occupation, construction worker (high physical demand); and comorbidity, not applicable. Patient-specific data of Figure 5B were as follows (white arrow head: chondral lesion; white arrow: complex tears and subluxation of medial meniscus): sex, male; age, 47; BMI, 26; occupation, farmer (high physical demand); comorbidity, not applicable.

## 4. Discussion

The principal findings of this study were as follows: several factors affecting the rate of remarkable progression and fate of primary OA were identified, and these were weighted by their degree of contribution to OA. Regarding the progression rate of OA, BMD, initial K–L grade, and physical occupational demands were major contributors, but initial K–L grade and physical occupational demands were more important contributors than BMD. Regarding the fate of OA, hypertension, initial K–L grade, and physical occupational demands were major contributors, but initial K–L grade and physical occupational demands were more significant than hypertension. The progression rate and fate of OA were more affected by the initial K–L grade and physical occupational demands (both were major contributors) than other factors. Furthermore, it was possible to predict the progression rate and fate of OA by considering patient-specific situations.

In this study, the proportion of men was lower (overall percentage of female patients is 81.4% (2028 of 2492)) than those of other previous studies [10,11]. The reasons for this disparity might be as follows. (1) The compliance of women was better in the inclusion and exclusion conditions of this study than men. (2) As is well known, elderly women have a higher rate of OA than men on account of physiological factors [10]. The proportion of 55 years ≤ age was bigger than that of age < 55 years (67% in overall and 74% in progression rate of OA). Finally, (3) there were are many situations in daily lifestyle and occupation to perform, e.g., kneel and squat in Korean women [11].

It is generally suggested that OA progression is associated with obesity, which is positively correlated with BMI [17,18]. However, in this study, a lower BMI had a greater effect on the progression rate and fate of OA. The relationship between BMD and OA has been controversial in many studies. Several studies have reported that a higher BMD is associated with a greater risk of developing knee OA [19,20,21,22]. However, in our study, patients with osteoporosis and osteopenia tended to have a faster rate of OA progression and a higher rate of surgical intervention. Aging and biological interactions may be possible reasons for these contradictory results, which may be associated with longitudinal cartilage loss in the knees [20,21,23]. Furthermore, low BMI can cause low BMD. Low BMD can cause microfracture at the bone under the cartilage or a varus deformity as a result of bowing. In addition, the varus deformity is common in Asians, and these factors are thought to have worked in combination [24,25].

The initial K–L grade was the most important factor affecting the progression and fate of OA. It is generally reported that varus alignment and chondro-meniscal status are strong risk factors for OA progression [12,26]. Osteotomy for patients with meniscal tears may present an opportunity for early intervention in patients with varus alignment and symptomatic early knee OA to limit the progression of OA and protect meniscal lesions [27,28]. This suggests that the treatment of OA must be started early and adapted according to the diagnosed K–L grade and other associated conditions of the knee, such as alignment and chondro-meniscal status accordingly.

In lifestyle, kneeling and squatting are considered the most important risk factors for knee disorders in many studies [6,26,29]. We classified physical occupational demands by modifying the reports of Palmer et al. and Reid et al. [14,16]. In our study, higher physical occupational demands were related to the progression rate and fate of OA, which corresponded well with results from previous reports [14,30]. Epidemiological studies have reported a positive association between OA and several comorbidities, such as hypertension and diabetes mellitus. Comorbidities are commonly associated with obesity, dyslipidemia, and hyperglycemia [12,26,31]. In this study, hypertension and diabetes mellitus tended to have a higher prevalence in patients with faster OA progression and earlier surgical intervention.

In this study, the hypertension and diabetes mellitus did not contribute to the progression rate and the fate of OA more than the initial K–L grade and physical occupational demands. This does not mean hypertension and diabetes mellitus were not related to OA. Hypertension and diabetes mellitus was also a contributor to the progression rate and the fate of OA. Each metabolic disease was already studied about its relationship with knee OA in some reports. In many studies, hypertension was associated with knee OA. Furthermore, hypertension was related to higher OA knee pain severity [32,33,34]. Our results also agree with those of studies. Hypertension was a major contributor to the fate of OA. It can be thought that hypertension is related to severe OA. Louati et al. reported a higher risk of knee OA significant exists in the diabetes mellitus population but causality was not yet clearly demonstrated [9]. However, some other meta-analyses do not support diabetes mellitus as an independent risk factor for knee OA. Rather, BMI was a more important confounding factor than diabetes mellitus [35,36]. Similarly, in our report, diabetes mellitus was a minor contributor to both the progression rate and the fate of OA.

Although many studies have attempted to develop a treatment plan for OA, decision making for the management of early OA is still ambiguous, even though the importance of the appropriate management of early OA has been emphasized. The reasons may be that many factors can affect the progression rate and fate of OA, and it might be difficult to identify the determining factors. CDWs provide serial data on various disease courses for OA, which makes it possible to analyze them [15,37]. The important difference between our method and those in previous reports is that our method leverages individualized patient-specific situations that are easy to obtain, such as occupation and comorbidities, and it tries to analyze them over time. In addition, we used patient-specific data acquired at a single institution, which could provide strong and consistent data to predict the progression rate and fate of OA. Our findings can be meaningful in supporting clinicians in planning treatment for patients with no radiologic or mild OA with knee pain using various methods, including pharmaceutical ones [38,39].

### Limitations

This study had several limitations. First, the proportion of men was lower compared to other previous studies [10,11]. However, it was uncontrollable and we could not find a specific reason that show this disparity. Secondly, we assumed K–L grade 2 and higher change as the same situation, because we focused on the rate for remarkable progression. In addition, our purpose was to find variables of remarkable progression for supporting clinicians in treatment planning. Thirdly, we studied a comprehensive analysis of patient-specific situations affecting the progress of OA and the fate of OA. Thus, this study can be limited in explaining the biological relationship between each factor and OA. Furthermore, in cases in which both knees were included (814 patients), there may be a lack of explanation about why the degree of OA in both knees differs. It might be the characteristics of the occupation and the differences of sides mainly used. Finally, each comorbidity included in the “other disorders” was too heterogeneous and their numbers were limited.

## 5. Conclusions

In identifying several contributing factors, the initial K–L grade and physical occupational demands were the most important factors. BMD and hypertension were also major contributors to the progression and fate of OA, and the degree of contribution was lower compared to the two major factors.

## Figures and Tables

**Figure 1 jcm-12-01204-f001:**
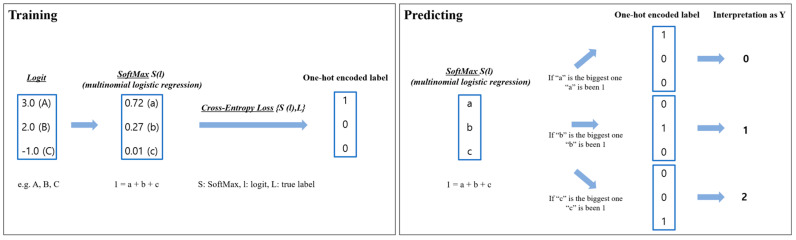
Schematic diagram of our model including training and predicting.

**Figure 2 jcm-12-01204-f002:**
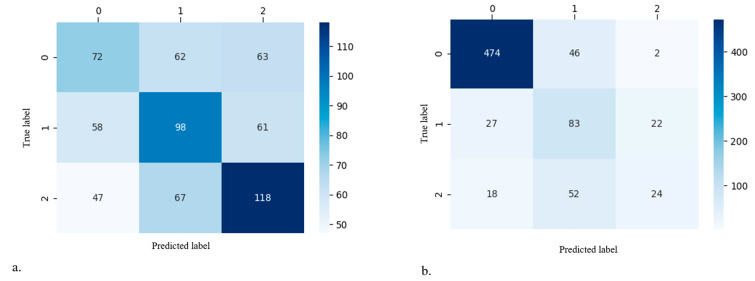
Confusion matrices of the best average precision model. (**a**) Progression rate of no or mild OA patients, and (**b**) fate of no or mild OA patients.

**Figure 3 jcm-12-01204-f003:**
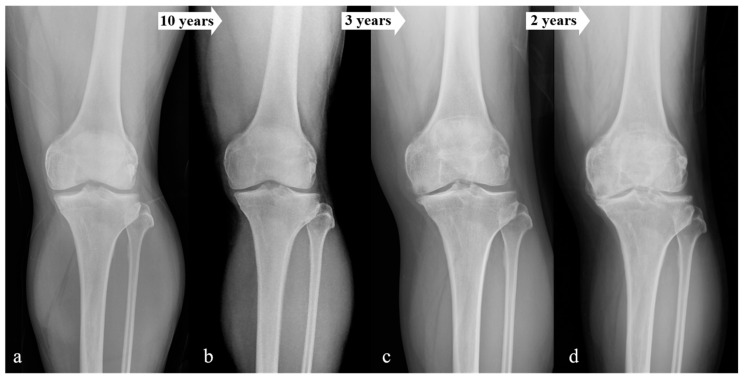
Serial changes of K–L grade, (**a**) initial anteroposterior (AP) X-ray (K–L grade 0-1); (**b**) 10 years later (K–L grade 2); (**c**) 13 years later (K–L grade 3); and (**d**) 15 years later (K–L grade 4).

**Figure 4 jcm-12-01204-f004:**
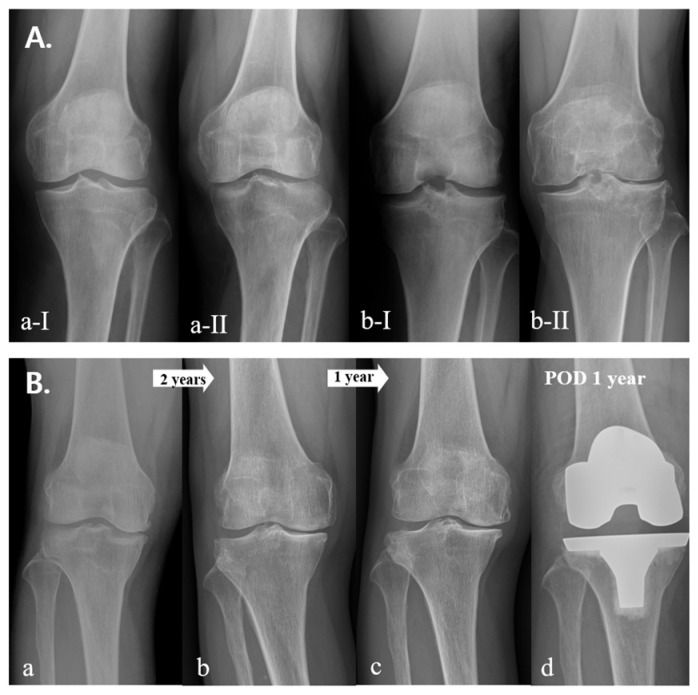
(**A**) Comparison of progression rate of OA according to initial K–L grade, (**a-I**) initial X-ray (K–L grade 0), (**a-II**) 10 years later (K–L grade 2), (**b-I**) initial X-ray (K–L grade 2), and (**b-II**) 3 years later (K–L grade 4). (**B**) Fate of rapid progression of severe OA, (**a**) initial X-ray (K–L grade 2); (**b**) 2 years later (K–L grade 3); (**c**) 3 years later K–L grade 4; and (**d**) 1 year after total knee arthroplasty.

**Figure 5 jcm-12-01204-f005:**
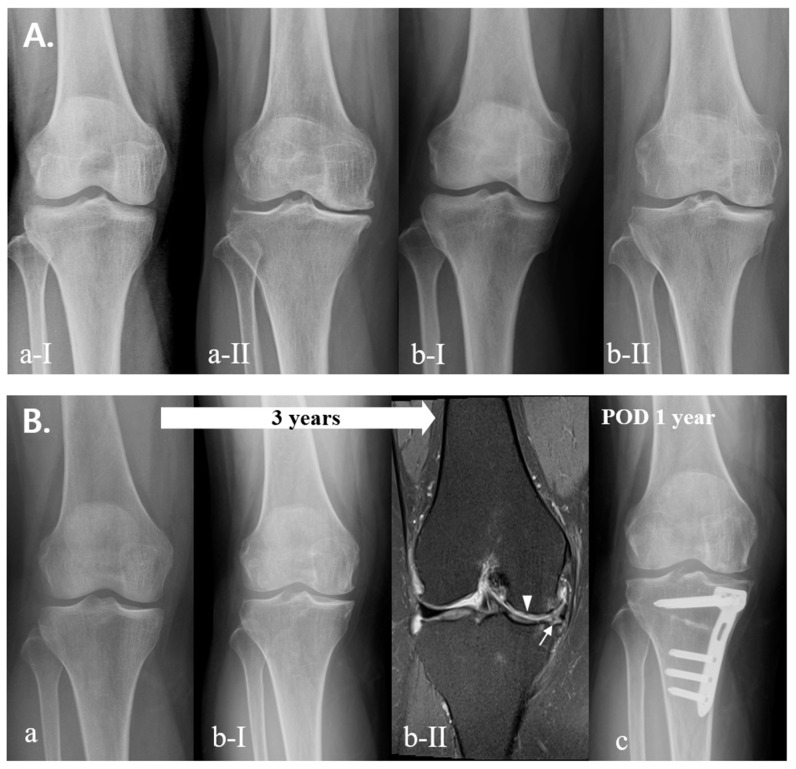
(**A**) Comparison of progression rate of OA according to physical occupational demands, (**a-I**) initial X-ray (K–L grade 1), and (**a-II**) 11 years later (K–L grade 3); (**b-I**) initial X-ray (K–L grade 1), and (**b-II**) 4 years later (K–L grade 3). (**B**) Fate of rapid progression of high physical demand worker, (**a**) initial X-ray (K–L grade 0); (**b-I**) 3 years later (K–L grade 2); (**b-II**) MRI of 3 years later; and (**c**) 1 year after high tibial osteotomy.

**Table 1 jcm-12-01204-t001:** Patient-specific situations and fate according to the initial K–L grade.

(Knees)	K–L Grade 0	K–L Grade 1	K–L Grade 2	*p*-Value
**Age** (Mean)	57.6 ± 11.9	59.7 ± 9.1	60.6 ± 8.1	<0.001 *
**Age < 55**	38.6% (714)	21.9% (66)	14.7% (50)	<0.001 *
**55 ≤ Age < 65**	31.9% (591)	48.0% (145)	60.2% (204)	
**Age ≥ 65**	29.5% (545)	28.1% (85)	27.1% (92)	
**Total** (2492)	100% (1850)	100% (303)	100% (339)	
**SEX** (Male/Female)	21.2%/78.8% (392/1458)	14.5%/85.5% (44/259)	8.3%/91.7% (28/311)	<0.001 *
**BMI**	25.3 ± 3.4	24.2 ± 3.2	23.0 ± 2.8	<0.001 *
**BMD**				0.001
**Normal** (159)	22.9% (116)	18.4% (21)	15.8% (22)	
**Osteopenia or Osteoporosis** (600)	77.1% (390)	81.6% (93)	85.2% (117)	
**Total performed** (759)	100% (506)	100% (114)	100% (139)	
**Physical demand for occupation**				<0.001 *
**Low demand** (813)	34.8% (643)	36.6% (111)	17.4% (59)	
**Mild demand** (1109)	51.5% (952)	23.4% (71)	25.4% (86)	
**High demand** (570)	13.8% (255)	39.9% (121)	57.2% (194)	
**Total** (2492)	100% (1850)	100% (303)	100% (339)	
**Rate of metabolic disorders**				
**HTN** (1204)	45.3% (838)	56.1% (170)	57.8% (196)	<0.001 *
**DM** (443)	17.5% (324)	21.1% (64)	16.2% (55)	0.227
**Other disorders** **(one or more)** (575)	21.8% (403)	26.1% (79)	27.4% (93)	0.032 *
**Fate**				
**Final K–L grade** (Mean)	2.3 ± 1.0	2.9 ± 0.9	3.8 ± 0.4	<0.001 *
**Surgical intervention**				<0.001 *
**Time < 5 years** (312)	8.3% (153)	13.9% (42)	34.5% (117)	
**Time ≥ 5 years** (439) (Max: 15 years)	10.3% (190)	27.4% (83)	49.0% (166)	
**Conservative treatment** (1741)	81.5% (1507)	58.7% (178)	16.5% (56)	
**Total** (2492)	100% (1850)	100% (303)	100% (339)	

OA, osteoarthritis; K–L, Kellgren–Lawrence; BMI, body mass index; BMD, bone mineral density; HTN, hypertension; DM, diabetes mellitus; Tb, tuberculosis. * Statistical significance was *p* < 0.05.

**Table 4 jcm-12-01204-t004:** Performance of model.

	Accuracy	Precision	Recall	F1-Score	Specificity	Error Rate
**Progression rate of OA**						
**Time < 5 years** (Fast)	0.632	0.488	0.509	0.498	0.700	0.368
**5 years ≤ Time < 10 years** (Usual)	0.616	0.432	0.452	0.441	0.699	0.384
**Time ≥ 10 years** (Slow)	0.644	0.407	0.365	0.385	0.766	0.356
**Fate of OA**						
**5 years > surgical intervention**	0.874	0.500	0.255	0.338	0.963	0.126
**5 years ≤ surgical intervention** (Max: 15 years)	0.803	0.459	0.629	0.530	0.841	0.197
**Conservative**	0.876	0.913	0.908	0.911	0.801	0.124

Time: elapsed K–L grade change more than 2; OA, osteoarthritis.

**Table 5 jcm-12-01204-t005:** Comparison of effect of each factor.

Coefficient (95% Confidence Interval)	Initial K–L Grade	Age	Sex	BMI	BMD	Physical Demand for Occupation	HTN	DM	Other Disorders (One or More)
**Progression rate of OA** (contribution)	Major *	Minor	Minor	Minor	Major *	Major *	Minor	Minor	Minor
**Time < 5 years** (Fast)	0.610 (0.609 ~ 0.610)	0.039 (0.039 ~ 0.039)	0.262 (0.262 ~ 0.262)	−0.099 (−0.099 ~ −0.098)	0.295 (0.294 ~ 0.295)	0.405 (0.404 ~ 0.405)	0.196 (0.196 ~ 0.197)	0.079 (0.079 ~ 0.079)	0.135 (0.135 ~ 0.136)
**5 years ≤ Time < 10 years** (Usual)	−0.125 (−0.126 ~ −0.125)	−0.012 (−0.013 ~ −0.012)	−0.092 (−0.092 ~ −0.092)	0.010 (0.010 ~ 0.011)	0.145 (0.145 ~ 0.145)	0.218 (0.217 ~ 0.218)	−0.075 (−0.075 ~ −0.075)	0.127 (0.127 ~ 0.128)	0.127 (0.126 ~ 0.127)
**Time ≥ 10 years** (Slow)	−0.667 (−0.668 ~ −0.667)	−0.027 (−0.027 ~ −0.027)	−0.178 (−0.179 ~ −0.178)	0.074 (0.074 ~ 0.075)	−0.459 (−0.460 ~ −0.459)	−0.669 (−0.669 ~ −0.669)	−0.087 (−0.088 ~ −0.087)	−0.078 (−0.079 ~ −0.078)	−0.088 (−0.088 ~ −0.088)
**Fate of OA (contribution)**	Major *	Minor	Minor	Minor	Minor	Major *	Major *	Minor	Minor
**5 years > surgical intervention**	0.462 (0.462 ~ 0.463)	0.131 (0.131 ~ 0.131)	0.271 (0.270 ~ 0.271)	−0.103 (−0.104 ~ −0.103)	0.225 (0.224 ~ 0.225)	0.496 (0.496 ~ 0.496)	0.290 (0.289 ~ 0.290)	0.239 (0.239 ~ 0.239)	−0.092 (−0.093 ~ 0.091)
**5 years ≤ surgical intervention** (Max: 15 years)	0.111 (0.110 ~ 0.111)	0.108 (0.108 ~ 0.108)	0.016 (0.015 ~ 0.017)	−0.030 (−0.031 ~ −0.030)	0.080 (0.079 ~ 0.080)	−0.156 (−0.156 ~ −0.156)	0.038 (0.038 ~ 0.038)	0.029 (0.029 ~ 0.029)	−0.085 (−0.086 ~ −0.085)
**Conservative**	−0.643 (−0.643 ~ −0.643)	−0.060 (−0.061 ~ −0.060)	−0.330 (−0.331 ~ −0.330)	0.071 (0.071 ~ 0.072)	−0.324 (−0.324 ~ −0.324)	−0.592 (−0.593 ~ −0.592)	−0.379 (−0.380 ~ −0.379)	−0.310 (−0.311 ~ −0.310)	0.339 (0.339 ~ 0.339)

K–L: Kellgren–Lawrence; time: elapsed K–L grade change more than 2; OA, osteoarthritis; BMI, body mass index; BMD, bone mineral density; HTN, hypertension; DM, diabetes mellitus; Tb, tuberculosis. * Average coefficient as major contributor ≥ |0.211|.

## Data Availability

Dr. Yong Seuk Lee (smcos1@hanmail.net) had full access to all data in the study and takes responsibility for the integrity of the data and accuracy of the data analysis.

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
