# Peer review of "Do Individualized Patient-Specific Situations Predict the Progression Rate and Fate of Knee Osteoarthritis? Prediction of Knee Osteoarthritis"

_jcm, 2023, doi:10.3390/jcm12031204_

Round 1

Reviewer 1 Report

This manuscript tried to identify patient specific information that could predict progression rate and fate of Osteoarthritis.

1. The authors stated that “2492 knees with pain and followed up for more than 1 year were analyzed”, since mostly one person has two knees, please provide information how many individuals were analyzed?  And the correlations between two knees for one person should be considered in your data analysis.

2. For your affection factor analysis using SoftMax, please clarify what is your final prediction model. Were those features in Tables 1, 2, and 3 all included in your final prediction model? Or just some of them were in your final model.

3. For table 5, please clarify, how you standardized these coefficients. You should clearly state how you standardized your coefficients for both categorical variables such as K-L grade and Sex and continuous variables such as BMI (there were no mentions of how this factor was treated in the manuscript).

4. I suggest you list all your coefficients for table 5 in an appendix file then readers can have a clearer view of the effects of different factors.

Author Response

Thank you for your constructive comments on our manuscript JCM (ISSN 2077-0383) Manuscript ID: jcm-2123084; Title: " Do Individualized Patient-specific Situations Predict the Progression Rate and Fate of Knee Osteoarthritis? Prediction of Knee Osteoarthritis"

We have tried our best to correct our manuscript as suggested by the reviewers. Specific details of the corrections are attached. Please find an attached file entitled ‘Response to reviewers’ letter. We have highlighted the revision file (track change) as you recommended and marked it as a yellow color.

< Point-by-point response to Evaluation >

Reviewer 1

This manuscript tried to identify patient specific information that could predict progression rate and fate of Osteoarthritis.

  1. The authors stated that “2492 knees with pain and followed up for more than 1 year were analyzed”, since mostly one person has two knees, please provide information how many individuals were analyzed?  And the correlations between two knees for one person should be considered in your data analysis.

Response: Thank you for your kind comments. As your comments, we added that information at line 87 and added limitations (Lines 341-343).

  1. For your affection factor analysis using SoftMax, please clarify what is your final prediction model. Were those features in Tables 1, 2, and 3 all included in your final prediction model? Or just some of them were in your final model.

Response: Thank you for your comments. We edited descriptions for more comprehensibility concerning your comments. (Lines 91-92, 111-112, 117-118, 142-143, and 155-156)

  1. For table 5, please clarify, how you standardized these coefficients. You should clearly state how you standardized your coefficients for both categorical variables such as K-L grade and Sex and continuous variables such as BMI (there were no mentions of how this factor was treated in the manuscript).

Response: Thank you for your comments. As your comments, we added a description at lines 129-130. The categorical variables were explained in lines 129-137.

  1. I suggest you list all your coefficients for table 5 in an appendix file then readers can have a clearer view of the effects of different factors.

Response: Thank you for your kindly comments. However, it would be enough that provides a 95% confidence interval, and we think it was concise to show as shown in Table 5.

Reviewer 2 Report

Yoo et al. aimed to identify if individualized patient-specific situations could predict the progression rate and fate of knee osteoarthritis (OA). Initially they analyzed data from 83280 patients with knee pain and finally they included 2492 knees with pain that were followed up for more than 1 year. They looked into patient-specific information that were categorized as demographic, radiologic, social, comorbidity disorders, and surgical intervention. They identified several factors affecting the rate of progression of primary OA, namely BMD, initial K-L grade, and physical occupational demand. Initial K-L grade and physical occupational demands were more important contributors than BMD. Major contributors of the fate of OA identified were hypertension, initial K-L grade, and physical occupational demands, but initial K-L grade and physical occupational demands were more significant than hypertension. The progression rate and fate of OA were more affected by the initial K-L grade and physical occupational demands than other factors. Based on these results, the authors conclude that is possible to predict the progression rate and fate of OA by considering patient-specific situations.

Since OA is a prevalent and debilitating disease lacking disease modifying drugs the results of the current study are of utmost importance for OA diagnosis and treatment. However, I believe it is limited to Asian ethnical group as some findings were contrary to what has been proved previously. The study is well designed and the manuscript is generally well written. I would only have some minor concerns pointed out below:

1.       The authors should use abbreviations such as OA, K-L score, DM, HTN etc. consistently, also in Abstract, meaning that when first mentioned, the word should be written out with its abbreviation in parentheses, and then the abbreviation should be consistently used. Please revise carefully the whole manuscript.

2.       Please also do not use arthritis instead of OA (line 260). Arthritis is more commonly referred to immunological rheumatoid arthritis.

3.       In Discussion (line 264 -272), the authors first discuss the connection between OA and obesity (BMI) and then with BMD. It is interesting that the authors found the opposite from what has been found in previous studies, meaning that OA progression is associated with higher BMI and BMD. Is this a specific situation for the Asian population as I assume this was the main ethnicity in your patient cohort?

Author Response

Thank you for your constructive comments on our manuscript JCM (ISSN 2077-0383) Manuscript ID: jcm-2123084; Title: " Do Individualized Patient-specific Situations Predict the Progression Rate and Fate of Knee Osteoarthritis? Prediction of Knee Osteoarthritis"

We have tried our best to correct our manuscript as suggested by the reviewers. Specific details of the corrections are attached. Please find an attached file entitled ‘Response to reviewers’ letter. We have highlighted the revision file (track change) as you recommended and marked it as a yellow color.

< Point-by-point response to Evaluation >

Reviewer 2

Yoo et al. aimed to identify if individualized patient-specific situations could predict the progression rate and fate of knee osteoarthritis (OA). Initially they analyzed data from 83280 patients with knee pain and finally they included 2492 knees with pain that were followed up for more than 1 year. They looked into patient-specific information that were categorized as demographic, radiologic, social, comorbidity disorders, and surgical intervention. They identified several factors affecting the rate of progression of primary OA, namely BMD, initial K-L grade, and physical occupational demand. Initial K-L grade and physical occupational demands were more important contributors than BMD. Major contributors of the fate of OA identified were hypertension, initial K-L grade, and physical occupational demands, but initial K-L grade and physical occupational demands were more significant than hypertension. The progression rate and fate of OA were more affected by the initial K-L grade and physical occupational demands than other factors. Based on these results, the authors conclude that is possible to predict the progression rate and fate of OA by considering patient-specific situations.

Since OA is a prevalent and debilitating disease lacking disease modifying drugs the results of the current study are of utmost importance for OA diagnosis and treatment. However, I believe it is limited to Asian ethnical group as some findings were contrary to what has been proved previously. The study is well designed and the manuscript is generally well written. I would only have some minor concerns pointed out below:

Response: Thank you for your kind comments.

  1. The authors should use abbreviations such as OA, K-L score, DM, HTN etc. consistently, also in Abstract, meaning that when first mentioned, the word should be written out with its abbreviation in parentheses, and then the abbreviation should be consistently used. Please revise carefully the whole manuscript.

 Response: Thank you for your comments. As your comments, we changed it at whole pages.

  1. Please also do not use arthritis instead of OA (line 260). Arthritis is more commonly referred to immunological rheumatoid arthritis.

Response: As your comments, we changed it at whole pages.

  1. In Discussion (line 264 -272), the authors first discuss the connection between OA and obesity (BMI) and then with BMD. It is interesting that the authors found the opposite from what has been found in previous studies, meaning that OA progression is associated with higher BMI and BMD. Is this a specific situation for the Asian population as I assume this was the main ethnicity in your patient cohort?

 Response: Thank you for your comments. We added an explanation and reference to your comments on lines 282-285.

Round 2

Reviewer 1 Report

Improved, no more comments.